# Identifying Hotspots of People Diagnosed of Tuberculosis with Addiction to Alcohol, Tobacco, and Other Drugs through a Geospatial Intelligence Application in Communities from Southern Brazil

**DOI:** 10.3390/tropicalmed7060082

**Published:** 2022-05-24

**Authors:** Alessandro Rolim Scholze, Felipe Mendes Delpino, Luana Seles Alves, Josilene Dália Alves, Thaís Zamboni Berra, Antônio Carlos Vieira Ramos, Miguel Fuentealba-Torres, Inês Fronteira, Ricardo Alexandre Arcêncio

**Affiliations:** 1Department of Maternal-Infant and Public Health Nursing, Ribeirão Preto College of Nursing, University of São Paulo, Ribeirão Preto 05403-000, Sao Paulo, Brazil; fmdsocial@outlook.com (F.M.D.); lu.selesrp@gmail.com (L.S.A.); thaiszamboni@live.com (T.Z.B.); antonio.vieiraramos@outlook.com (A.C.V.R.); ricardo@eerp.usp.br (R.A.A.); 2Institute of Biological Sciences and Health, Federal University of Mato Grosso, Barra do Garças 78600-000, Mato Grosso, Brazil; josydalia@hotmail.com; 3Nursing Department, Faculdade de Enfermagem e Obstetrícia, Universidad de los Andes, Santiago 12455, Chile; mafuentealba@uandes.cl; 4Global Health and Tropical Medicine, Instituto de Higiene e Medicina Tropical, Universidade Nova de Lisboa, 1349-008 Lisbon, Portugal; ifronteira@ihmt.unl.pt

**Keywords:** tuberculosis, alcoholism, illicit drugs, tobacco use disorder, spatial analysis

## Abstract

(1) Background: tuberculosis (TB) is considered one of the leading causes of death worldwide by a single infectious agent. This study aimed to identify hotspots of people diagnosed with tuberculosis and abusive use of alcohol, tobacco, and other drugs in communities through a geospatial intelligence application; (2) Methods: an ecological study with a spatio-temporal approach. We considered tuberculosis cases diagnosed and registered in the Notifiable Diseases Information System, which presented information on alcoholism, smoking, and drug abuse. Spatial Variations in Temporal Trends (SVTT) and scan statistics were applied for the identification of Hotspots; (3) Results: between the study period, about 29,499 cases of tuberculosis were reported. When we applied the SVTT for alcoholism, three Hotspots were detected, one of which was protective (RR: 0.08–CI95%: 0.02–0.32) and two at risk (RR: 1.42–CI95%: 1.11–1.73; RR: 1.39–CI95%: 1.28–1.50). Regarding smoking, two risk clusters were identified (RR: 1.15–CI95%: 1.01–1.30; RR: 1.68–CI95%: 1.54–1.83). For other drugs, a risk cluster was found (RR: 1.13–CI95%: 0.99–1.29) and two protections (RR: 0.70–CI95%: 0.63–0.77; RR: 0.76–CI95%: 0.65–0.89); (4) Conclusion: it was evidenced that in the communities being studied, there exists a problem of TB with drug addiction. The disordered use of these substances may harm a person’s brain and behavior and lead to an inability to continue their treatment, putting the community at further risk for TB.

## 1. Introduction

Tuberculosis (TB) is considered one of the leading causes of death by a single infectious agent worldwide. It is one of the leading causes of preventable death that persists across the population [1,2,3]. Brazil has been one of the countries with the highest number of TB cases since 2003, ranking 16th position among the 22 countries with the highest TB cases in the world. These countries are responsible for a global burden of the disease (nearly 80%) [2,3,4].

In 2020, Brazil presented an incidence of 31.6 cases per 100,000 inhabitants [3]. It is noteworthy that many factors influence the sustenance of the disease burden in the country, such as social inequalities that significantly increase the burden of the disease and lead to unfavorable outcomes, including treatment dropout, hospitalizations, drug-resistant TB (DR-TB), and death [5,6].

TB affects vulnerable populations, mainly those with mental health disorders and substance use disorders, such as alcohol use, tobacco, and other drugs [7,8]. A study conducted in South Africa showed an estimated 10.4 million new cases of TB, of which 4.7% were related to harmful alcohol use [9]. Thus, the use of alcohol, tobacco, and other drugs should be considered in the elaboration of measures and strategies aimed at eliminating TB [8].

Studies [1,7,8,9,10] suggested an association between alcohol, tobacco, and other drugs with latent infection among people who make chronic use of these substances. Those people have greater odds of the development of active TB, treatment failures related to therapeutic interruptions and/or discontinuities, and early deaths, mainly in the male population.

Although the association between TB and alcohol, tobacco, and other drugs is a priority issue to design health policies addressed to eliminate the disease, it is still underexplored in the scientific literature from the perspective of vulnerable populations and hazard-prone territories. This study aligns with the National Plan to End Tuberculosis, which includes the need to intensify strategic actions aimed at the populations most vulnerable to tuberculosis [3]. Thus, the fundamental importance of this study for the scientific community is justified since there is a large gap of knowledge directed at vulnerable populations, as well as when TB is associated with users of alcohol, tobacco, and other drugs.

In Brazil, although each territory is under the accountability of health services and health workers, most of them do not have appropriate technologies for stratifying the risk that their community is exposed to, which makes the geospatial intelligence an important ally [11]. Therefore, the study aimed to identify Hotspots of people diagnosed with tuberculosis with an abusive use of alcohol, tobacco, and other drugs in communities through a geospatial intelligence application in Southern Brazil.

## 2. Materials and Methods

### 2.1. Study Design and Location

An ecological study conducted in the state of Paraná, located in the South of Brazil macro-region and located in the geographical coordinates 24°59′ S latitude and 53°56′ W longitude, with an estimated population of 11.34 million inhabitants, the fifteenth state which has the largest national territory and the fifth-largest population [5]. 

For the study, as a unit of geographical analysis we used the 399 municipalities that make up the state of Paraná, which is subdivided into 22 regional health sectors and nine mesoregional municipalities: West, Northwest, Western Center, Central North, Pioneer North, Eastern Center, Metropolitan of Curitiba, Southeast, Center South, and Southwest. Figure 1 illustrates the location of the state and its mesoregions.

### 2.2. Population, Source of Information, and Study Variables

The study used secondary data from TB notifications registered in the Information System of Notifiable Diseases (SINAN), of Paraná, from 2008–2018. These data were made available by the Secretary of State for Health (SESA) in an Excel file on 11 May 2020. 

Confirmed TB cases were considered in the study population and those who had a self-report of alcohol, tobacco, and other illicit drug use while filling out the notification form. Cases that did not contain the use of alcohol, tobacco, and other drugs and missing data were excluded.

We used the following variables available in SINAN: sex: (male and female); age group: (<15 years, 15–29 years, 30–59 years, and >60 years); race: (white, black, yellow, brown and indigenous); schooling: (illiterate, up to eight years of study and more than eight years of study); type of case: (new case, recurrence, reinstatement after dropout, and transfer); classification: (pulmonary, extrapulmonary, and pulmonary + extrapulmonary); drug sensitivity testing (resistant only to isoniazid, resistant to rifampicin, resistant to isoniazid and rifampicin, resistant to other 1st line drugs, and sensitive); outcome of the treatment (cure, abandonment, death from TB, death from other causes, and DR-TB).

### 2.3. Data Analysis

Initially, descriptive statistics were applied to calculate absolute and relative frequency measurements for categorical variables using the statistical software R Studio® version 3.5.2, R Core Team (2020). R: A language and environment for statistical computing. R Foundation for Statistical.

Computing, Vienna, Austria. URL https://www.R-project.org/, accessed on 18 May 2022.

To identify Hotspots, the geographic coordinates of each municipality were first obtained from the free access Google Earth tool. Then the case georeferencing technique was performed with the ArcGIS [GIS software]. Version 10.0. Redlands, CA, USA: Environmental Systems Research Institute, Inc., 2010.

Then, for the detection of clusters of spatial risk for the occurrence of TB concomitantly with the use of alcohol, tobacco, and other drugs, we used the Purely Spatial Scan Statistics (EVPP), developed by Kulldorff and Nagarwalla (1995) [12], and used the Satscan software (version 9.6). Department of Medicine. Harvard Medical School. Tremont Street, 3rd Floor, Boston, MA, USA.

The scanning statistics consists of the formation of circles that move throughout the area under study, that is, the state of Paraná, around the centroids, which correspond to the center of each municipality under analysis; a centralization process is performed, the radius of which may vary from zero to the limit determined by the researcher [13].

The identification of Hotspots was performed by calculating the number of events found within each circle. If the observed value was higher than expected in the region *z* de-limited by the circle called agglomerate, the radius of the circle was enlarged to a new centroid, and this occurred until all centroids were tested under the following hypotheses: H_0_: no agglomeration in the study region and H_1_: the region z is a cluster [14], so that cluster was understood to be the region with greater or lower risk of having TB and using alcohol, tobacco or other drugs when compared to other regions.

The Likelihood Function calculated to define Hotspots has been maximized in general in the checked windows; the maximum log-likelihood ratio (Likelihood Function - LLR) corresponds to the most likely cluster, which means it is the least likely to have occurred at random; additionally, other statistically significant ordinal LLRs have been combined with secondary clusters.

The *p*-value of the maximum likelihood test was obtained through Monte Carlo hypothesis tests and randomly-replicated simulation tests to compare the maximum LLR classification of real data with random data.

The Hotspot’s relative risk (RR) was defined as the risk within the scan window compared to the risk outside the scan window, representing how more common the disease is in a given location and time period than the baseline [15]. RR equal to 1 means no statistically significant difference between exposed and unexposed groups; RR < 1 is equivalent to low risk (or protection), RR > 1 can be understood as an area of risk.

The following criteria were used to identify Hotspots: Poisson Discrete Model, non-overlapping geographic clusters, maximum cluster size equal to 50% of the exposed population, agglomerates with a circular shape, and 999 replications following the Monte Carlo criteria.

After the identification of the purely spatial Hotspot [16] and temporal space, to assess the reliability of RR values, the respective 95% confidence intervals (95%CI) were calculated. Thematic maps were developed using ArcGIS [GIS software]. Version 10.0. Redlands, CA, USA, EUA: Environmental Systems Research Institute, Inc., 2010.

Then, the Spatial Variation in Temporal Trends (SVTT) technique was used to detect and infer risk Hotspots with significantly different time trends. In this analysis, the scan window is purely spatial in nature [16]. However, the time trend is calculated inside and outside the scan window for each location and size.

When a difference in temporal trend between internal and external areas is detected, its statistical significance is calculated [17]. The following hypotheses were tested in each window: H_0_: the time trends are the same in all areas, and H_1_: the trends are different. Thus, risk clusters indicate areas with a statistically different temporal trend from the temporal trend outside these clusters (considering a type I error of 5%).

The results indicate an Internal Temporal Trend (ITT), which consists of the degree of growth or decrease of the event within the cluster of risk, and an External Temporal Trend (ETT), which corresponds to the trend of all other areas which do not belong to this cluster in question. It is valid to highlight that the ITT and ETT found in this area were statistically significant in this analysis and not the cluster itself. Therefore, the 95%CI was not calculated for this analysis [18].

## 3. Results

From 2008–2018, 29,499 TB cases were reported in the state of Paraná, of which 32.41% (n = 9.561) used some type of psychoactive substance, with alcoholism being 45.78% (n = 6.014), smoking 32.09% (n = 4.216), and other drug use 22.13% (n = 2.903).

When analyzing the sociodemographic characteristics of TB cases among users, we observed a higher prevalence of male sex, age from 30 to 59 years, race-white color, and up to eight years of study (Table 1).

Regarding the clinical characteristics of TB cases described in Table 1, the majority corresponded to new and pulmonary cases and were sensitive to the available medication. As for the outcome, most cases evolved to cure, and it was still observed that 11.67% had abandoned treatment.

In Figure 2, we observed 6.014 cases of TB and alcoholism with an incidence rate of 5.4 cases/100,000 inhabitants. According to the analysis result, an annual growth of 0.58% was identified in Paraná. From the EVPP analysis, it is possible to observe the conformation of eight clusters of spatial risk, elucidated in Figure 2A, in the West, Northwest, Pioneer North, and the Metropolitan of Curitiba.

Figure 2B shows the Hotspots obtained from the SVTT analysis for TB, in which it is possible to observe a protective cluster in the Northwest *Paranaense* mesoregion with ITT above 99% and ETT = 0.57% growth per year. However, on the SVTT, two risk clusters were observed, with Cluster 2 in the North Central mesoregion, with ITT = 50.51% annual growth and ETT = 0.30% growth per year, and Cluster 3 in the Metropolitan region of Curitiba, with ITT = 6.6% and TTE = 0.20%; both showed annual growth.

There were 4.216 cases of TB and smoking, whose incidence rate was 3.8 cases/100,000 inhabitants, and there was an annual growth of 37.08% in the state in the Sof Stats (Figure 3). The EVPP identified seven clusters of spatial risk in the West, Northwest, Central, and Metropolitan North regions of Curitiba (Figure 3A).

The SVTT for this condition pointed to two risk Hotspots, with Cluster 1 extending from the West to the Northwest mesoregion, with ITT = 70.8 annual growth and ETT = 35.7% growth per year, and Cluster 2 in the Metropolitan region of Curitiba with LLR= 21.6 ITT = 52.7% and ETT = 34.8% annual growth (Figure 3B).

Regarding TB cases and the use of other drugs, we observed 2.903 cases, and the incidence rate was 2.6 cases/100,000 inhabitants, with a 19.41% annual growth in the state (Figure 4). Figure 4A represents the EVPP, which identified five clusters of spatial risk in the West, North Central, and Metropolitan regions of Curitiba.

As shown in Figure 4B, the SVTT analysis for TB and other drugs identified two protection clusters and one at-risk cluster. The protection Cluster 1 reached the entire length of the Western, Northwest, Western Center, South, and Southwest mesoregion, with ITT = 36.7% of annual growth and ETT = 16.3% of growth per year, and Cluster 3 in the Metropolitan mesoregion of Curitiba, ITT of 37.3% and ETT of 18.5% annual growth. Cluster 2 of risk is located on the boundary between the Pioneer North and Central Oriental mesoregions, with ITT of 70.8% annual growth and ETT of 35.7% of growth per year.

## 4. Discussion

The study aimed to identify Hotspots of people diagnosed with tuberculosis and abusive use of alcohol, tobacco, and other drugs in communities through a geospatial intelligence application. According to the results, Hotspots were identified according to the type of drug used by people with TB; this may be associated with the social determinants of the different regions and/or territories identified.

We found spatial variations in the temporal trends of alcohol, tobacco, and other drugs, showing that the phenomenon is growing in this territory. The incidence of TB cases associated with alcohol, tobacco, and other drugs was also verified, with a heterogeneous distribution between the state’s different regions.

TB is historically known for its direct relationship with social determinants in health and for reaching the most vulnerable population groups, a factor that requires an individualized, collective, and programmatic health strategy for its elimination [17].

Among the different risk factors that hinder/favor the development of TB, alcohol stands out. Alcohol consumption is an ancient practice and is part of the socio-historical process of humanity. However, in excess, this habit can generate negative consequences, such as the chemical-dependence development of more than 60 types of acute and chronic diseases, including TB [19].

The use of alcohol/alcoholism is a determinant with a strong association with the development of TB, a condition identified in this study, among 45.78% of the population with TB. Studies developed in Singapore [20], Germany [21], Ethiopia [22], India [23], and Brazil [6,10,24] presented the same specificities, alcoholism being a risk factor for TB and responsible for worse treatment outcomes. 

Alcohol consumption is a negative factor for the prognosis of TB since it triggers difficulty in treatment adherence, interruptions in the use of medicines, and deaths [9]. When an individual does not adhere to the treatment of TB, this contributes significantly to the increase in the number of new cases in geographical space [24]. Therefore, the consumption of alcohol and other drugs can aggravate the non-adherence to treatment, the transmission of the disease, and cause death.

It is also noteworthy that the alcohol consumption disorder can result in greater chances for liver damage, altering the metabolism of antibacterial drugs and that alcohol is a significant predictor of the worst outcomes in TB treatment. This factor contributes to the development of TB drug resistance [25].

In a randomized clinical trial developed in India [23], which sought to demonstrate the feasibility of using intervention regarding alcohol consumption among patients with TB, it was evidenced that in the intervention group, the results were significantly higher (*p* = 0.02) when treatment adherence was identified.

In a systematic review study with meta-analysis, the estimated mean prevalence of alcohol use disorder in patients with TB was 30% (95% CI: 24.00, 35.00). The countries with the highest prevalence were Asia and Europe at 37%. When analyzing the profile of patients, TB was higher among patients aged over 40 years (42%), male sex, being single, unemployed, low schooling and income (poverty), retraction, and treatment failure [25].

This study shows a relationship between TB and smoking, as 14.29% of the reported cases were smokers. It is estimated that about 1.3 billion people worldwide make active use of tobacco, with a higher incidence in underdeveloped or/in developed countries. Consequently, these countries have a high TB rate [26].

Tobacco is also understood as an important risk factor for TB, and exposure to cigarette smoke increases three times the chances of developing TB [27]. Among smokers, the risk of latent TB increases 1.6 times. Among active TB cases, the risk is two times higher for developing the disease and the chance of death is 2.6 times higher. It is also worth mentioning that the chances for development and worsening/death increase significantly with the time of consumption in years, the number of cigarettes used per day, and the socioeconomic level [28,29]. Tobacco use is a more common preventable risk factor for mortality and premature morbidity [30].

A study showed that when developing policies and actions aimed at abstinence from smoking among patients diagnosed with TB, there was a decrease in the rate of diagnoses and deaths after smoking cessation [31]. Thus, after six weeks of smoking cessation, the adverse effects of tobacco on the immune system disappear and contribute to a better clinical evolution of the patient [32].

Thus, it is suggested that health actions aimed at smoking cessation be implemented among patients diagnosed with TB. In this sense, these activities can be implemented in clinical and non-clinical environments. In clinical environments, activities aimed at identifying current and past smoking, routine counseling of patients seen at the clinic, offering intensive support with counseling of tobacco users, and administration of nicotine-replacement therapy for some patients. In non-clinical environments, they include educating the population about the dangers of tobacco use and passive smoking [33].

A systematic review pointed out that among facilitators to assist in smoking cessation are repeated brief interventions (including motivational therapy and leaflet distribution), followed by behavioral counseling incorporated into the routine of health care for patients with TB [31].

In a study that sought to analyze the risk ratios for the effect of smoking on TB incidence and mortality, the authors observed that 17.6% (95% confidence interval (CI): 8.4, 21.4) of TB cases and 15.2% (95% CI: 1.8, 31.9) of TB mortality were attributable to smoking [34].

Among the cases of TB in the state, 22.13% were users of other drugs. Knowing the direct relationship between TB and social determinants of health, it is known that the population using illicit and injectable drugs has a greater vulnerability to TB infection and a worse outcome in treatment, and often these users find an environment of marginalization in the streets and lack access to health services [24].

The use of illicit drugs causes numerous problems in the health of the population, decreases the body’s immune response and consequently increases the likelihood of the development of TB. Those diagnosed with the disease have a higher bacillary burden and likelihood of developing drug resistance [35], related to lack of demand/access to health services and difficulty performing the recommended TB treatment. Expansion of the office’s teams in the street and DOTS for these people may reduce illicit drug users’ adverse health, social, and economic consequences without necessarily reducing consumption [29,35].

Another aggravating factor among illicit drug users is the use of injectable drugs, these individuals are considered a group with a high risk for TB and are more susceptible to infection by the human immunodeficiency virus. This condition directly alters the immune response mediated by the characteristic body defense cells, which contribute to the worst outcome in the treatment of the disease [36].

Spatial clusters are multicausal, and it is important to understand that they are neither stagnant nor homogeneous; within the spatial cluster itself, there is a difference in relative risk, both decrease, and increase; therefore, it is important to use more than one methodology to be able to characterize the agglomeration.

The originality of this study is allowing health managers and workers to direct specific actions to the population who is addicted to alcohol and other drugs, as well as knowing the epidemiological scenario and developing strategies to promote a reduction in the TB transmission cycle and development of new cases [37], as well as treatment abandonment, death, and multidrug-resistant TB [24].

The main limitation of this study includes the use of secondary data, which is not free of incomplete data. However, Sinan is considered a reliable source for developing studies of this nature.

## 5. Conclusions

The study advances knowledge by evidencing the most vulnerable territories to the problem, as TB comorbidity and the use of alcohol and other drugs have been confirmed over time in these territories. This evidence gives rise to reflections on the measures implemented in these regions and how much they have had an effect.

The study brings evidence currently available to the panels defined by the Ministry of Science and Technology to adopt and define public policies. It is also expected that the results are reversed in updating projects of the teams that are currently on the front line of the fight against TB so that they recognize themselves in the process and can think of more comprehensive work projects that include mental health, as a possibility of joint work and intersectoral actions. The elimination of TB is based on the United Nations (UN) Sustainable Development Goals and the End TB Strategy.

## Figures and Tables

**Figure 1 tropicalmed-07-00082-f001:**
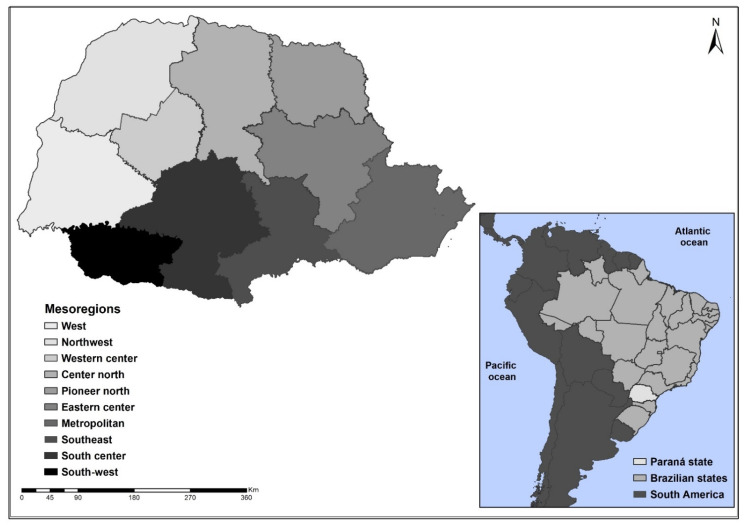
Geographic location of the state of Paraná and its mesoregions. Legend: ArcGIS [GIS software]. Version 10.0. Redlands, CA, USA, EUA: Environmental Systems Research Institute, Inc., 2010.

**Figure 2 tropicalmed-07-00082-f002:**
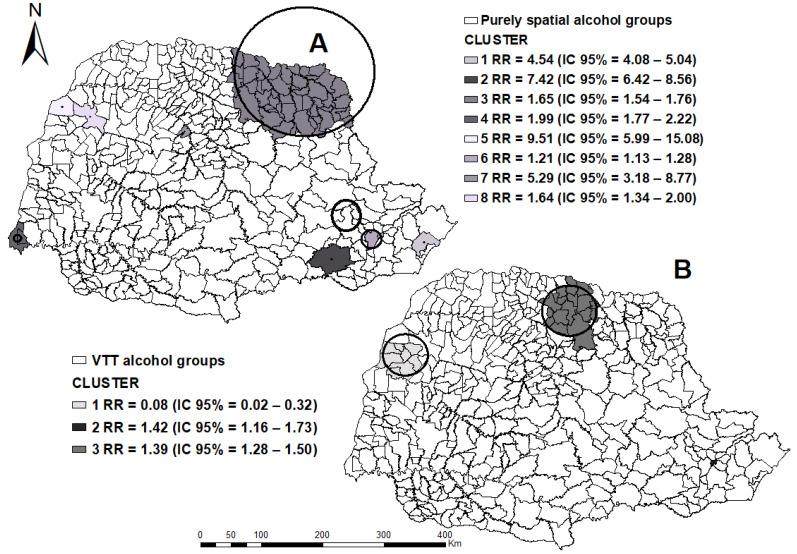
Areas of spatial risk for tuberculosis in alcohol users, by municipalities, state of Paraná (2008–2018). (**A**) applied the purely spatial technique; (**B**) applied the purely Spatial Variation in Temporal Trends. Legend: ArcGIS [GIS software]. Version 10.0. Redlands, CA, USA, EUA: Environmental Systems Research Institute, Inc., 2010.

**Figure 3 tropicalmed-07-00082-f003:**
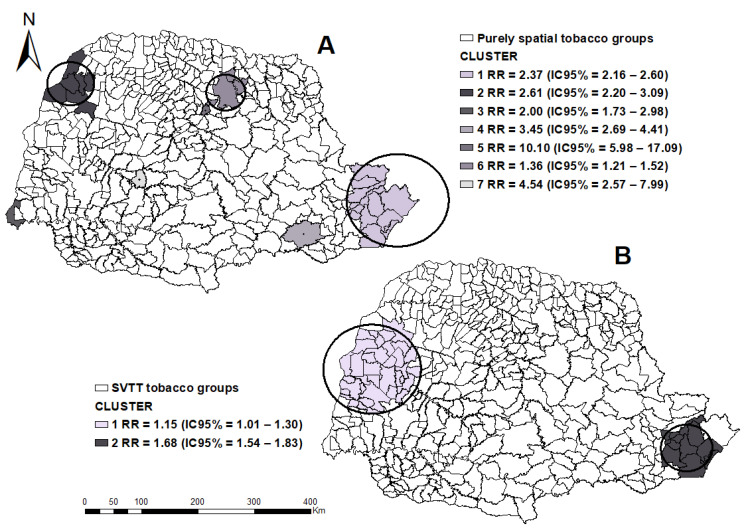
Areas at spatial risk for tuberculosis in tobacco users, by municipalities, state of Paraná (2008–2018). (**A**) applied the purely spatial technique; (**B**) applied the purely Spatial Variation in Temporal Trends. Legend: ArcGIS [GIS software]. Version 10.0. Redlands, CA, USA, EUA: Environmental Systems Research Institute, Inc., 2010.

**Figure 4 tropicalmed-07-00082-f004:**
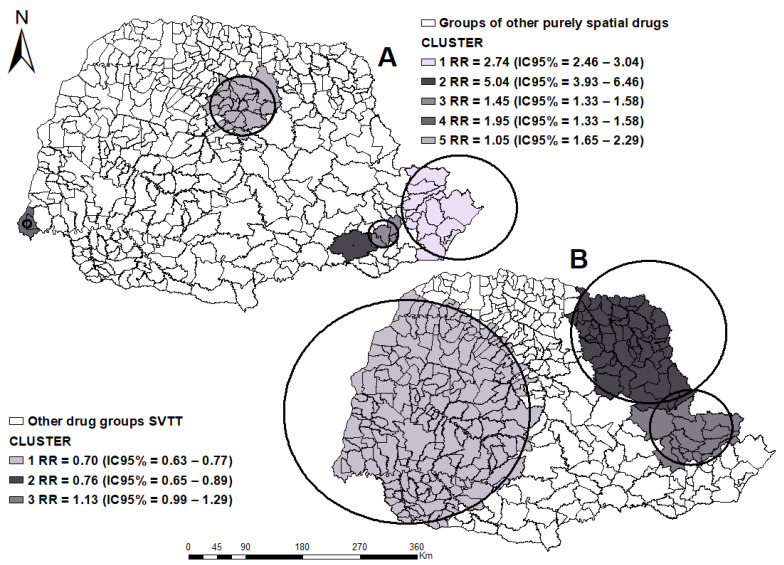
Areas of spatial risk for tuberculosis in users of other drugs, by municipalities, state of Paraná (2008–2018). (**A**) applied the purely spatial technique; (**B**) applied the purely Spatial Variation in Temporal Trends. Legend: ArcGIS [GIS software]. Version 10.0. Redlands, CA, USA, EUA: Environmental Systems Research Institute, Inc., 2010.

**Table 1 tropicalmed-07-00082-t001:** Clinical characteristics of TB patients, Paraná, Brazil (2008–2018).

Variables	Absolute Frequency (n = 9561)	Relative Frequency (%)	Rate (100,000 Inhabitants)
Sex			
Male	7723	80.78	80,776.07
Female	1838	19.22	19,223.93
Age group			
<15 years	83	0.87	868.11
15–29 years	1994	20.86	20,855.56
30–59 years	6517	68.16	68,162.33
>60 years	967	10.11	10,114.00
Race			
White	5750	60.14	60,140.15
Black	848	8.87	8869.37
Yellow	84	0.88	878.57
Brown	2643	27.64	27,643.55
Indigenous	52	0.54	543.88
Schooling			
Illiterate	375	3.92	3922.18
Up to 8 years of study	6.461	67.60	18,899.70
More than 8 years of study	2.345	24.50	10,720.64
Case type			
New case	7658	80.1	80,096.22
Recurrence	689	7.21	7206.36
Reinstatement after abandonment	612	6.4	6401
Transfer	555	5.8	5804.83
Classification			
Pulmonary	8307	86.88	86,884.22
Extrapulmonary	977	10.22	10,218.6
Pulmonary + extrapulmonary	277	2.9	2897.19
Sensitivity test			
Resistant to isoniazid only	112	1.17	1171.43
Resistant to rifampicin	16	0.17	167.35
Resistant to isoniazid and rifampicin	29	0.3	303.32
Resistant to other 1st-line drugs	35	0.37	366.07
Sensible	1053	11.01	11,013.49
Closing status			
Cure	6408	67.02	67,022.28
Abandonment	1116	11.67	11,672.42
Death from TB	451	4.72	4717.08
Death from other causes	621	6.5	6495.14
DR-TB	212	2.22	2217.34

## Data Availability

The data presented in this study are available on request from the corresponding author on reasonable request.

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
