# Peer review of "Identifying Hotspots of People Diagnosed of Tuberculosis with Addiction to Alcohol, Tobacco, and Other Drugs through a Geospatial Intelligence Application in Communities from Southern Brazil"

_tropicalmed, 2022, doi:10.3390/tropicalmed7060082_

Round 1
Reviewer 1 Report
In this study, Rolim Scholze A J et al. had the aim to identify hotspots of people diagnosed with tuberculosis and the use of alcohol, tobacco, and other drugs in communities through a geo-spatial intelligence application. The authors applied spatial Variations in Temporal Trends (SVTT) and scan statistics and concluded that the communities (Southern Brazil) have a problem with TB/ drug addiction. The last phrase of the conclusion is not clear to me.
In my viewpoint, this manuscript was well planned, used adequate methods, and provided beneficial information that could be helpful to design health policies addressed to eliminate the disease. Minor changes are required to improve the manuscript.
Minor comments
- The last phrase of the conclusion is not precise (abstract); please re-phrase it.
- Secretariat and Secretary (lines 95 and 96, respectively) mean the same?
- Please include the software where the figures were made in the figure legend.
- The manuscript has some grammatical mistakes.
Author Response
In this study, Rolim Scholze A J et al. had the aim to identify hotspots of people diagnosed with tuberculosis and the use of alcohol, tobacco, and other drugs in communities through a geo-spatial intelligence application. The authors applied spatial Variations in Temporal Trends (SVTT) and scan statistics and concluded that the communities (Southern Brazil) have a problem with TB/ drug addiction. The last phrase of the conclusion is not clear to me.
In my viewpoint, this manuscript was well planned, used adequate methods, and provided beneficial information that could be helpful to design health policies addressed to eliminate the disease. Minor changes are required to improve the manuscript.
Dear reviewer, we appreciate your contributions. Below are the answers to your comments, which were fundamental to improving the manuscript.
Minor comments
- The last phrase of the conclusion is not precise (abstract); please re-phrase it.
Response 1- We just excluded it, thank you.
- Secretariat and Secretary (lines 95 and 96, respectively) mean the same?
Response 2- Adequacy was made in writing "notifications registered in the Information System of Notifiable Diseases (SINAN), Paraná.
- Please include the software where the figures were made in the figure legend.
Response 3- It was included, thanks for the suggestion.
- The manuscript has some grammatical mistakes.
Response 4- A grammar review was performed, Thank you.

Reviewer 2 Report
The first sentence in Results subheading within the Abstract should be modified, it can not start with " of 29.499 cases...".
In the whole manuscript there should be space between word and bracket with reference number for example lines 49, 52, 53 etc.
The Discussion is rather poor, so I recommend more comparisons with available data from other similar studies in different countries that used this or similar technological solutions.
Author Response
Dear reviewer, we appreciate your contributions. Below are the answers to your comments, which were fundamental to improving the manuscript.
1- The first sentence in Results subheading within the Abstract should be modified, it can not start with " of 29.499 cases...".
Response 1- It was modified, thank you.
2- In the whole manuscript there should be space between word and bracket with reference number for example lines 49, 52, 53 etc.
Response 2- It was adjusted; sorry for the mistake.
3- The Discussion is rather poor, so I recommend more comparisons with available data from other similar studies in different countries that used this or similar technological solutions.
Response 3- As requested, we made a further deepening in the discussion. We appreciate the suggestion.

Reviewer 3 Report
General remark.
The objectives of the article are particularly relevant.
The methods by which the hotspot are created are not straightforward and should be more carefully described in detail.
The overall organisation of the article should be rearranged. The background should be placed at the beginning.
Conclusions do not clarify how such data could be used to avoid new cases of TB
Specific comments
Page 1, lines 34 and 76: Readers would undoubtedly appreciate explaining the meaning of hotspot briefly (i.e. a place with a significant hazard/risk).
Fig 1. Please, whenever possible, please use the English language
Pag 3 lines 931-109 It is not clear to me how did the Authors retrieve the data dealing with alcohol, drugs and so on
Pag 4 lines 114-119: Such a detailed description of a “rate” is unnecessary.
Pag 5 line 176-181: Ethical aspects are usually mentioned at the end of the article.
Pag 5 lines 182-184: see note pag 3 (931-109)
Pag 10 lines 248-270; 275-281; such a section would be more appropriately placed in the background. At the beginning of the article
Author Response
Dear reviewer, we appreciate your contributions. Below are the answers to your comments, which were fundamental to improving the manuscript.
General remark.
The objectives of the article are particularly relevant.
Response- Thank you
The methods by which the hotspot are created are not straightforward and should be more carefully described in detail.
Response- We just included a paragraph (marked in red). Thanks for the suggestion.
The overall organisation of the article should be rearranged. The background should be placed at the beginning.
Response- Some parts have been reorganized, and the ones we kept have answered all the questions. Thank you for the suggestion.
Conclusions do not clarify how such data could be used to avoid new cases of TB
Response- Readjusted and deleted the final sentence as requested
Specific comments
Page 1, lines 34 and 76: Readers would undoubtedly appreciate explaining the meaning of hotspot briefly (i.e. a place with a significant hazard/risk).
Response- We just included a paragraph (marked in red). Thanks for the suggestion.
Fig 1. Please, whenever possible, please use the English language
Response- We just updated to English.
Pag 3 lines 931-109 It is not clear to me how did the Authors retrieve the data dealing with alcohol, drugs and so on
Response- The TB notification form has different investigation variables between these variables and those related to the consumption of alcohol, tobacco, and illicit drugs. In this sense, for the development of this study, selected these variables.
Pag 4 lines 114-119: Such a detailed description of a “rate” is unnecessary.
Response- It was excluded, thank you.
Pag 5 line 176-181: Ethical aspects are usually mentioned at the end of the article.
Response- We just included. Thanks for the suggestion.
Pag 5 lines 182-184: see note pag 3 (931-109)
Response- It was excluded.
Pag 10 lines 248-270; 275-281; such a section would be more appropriately placed in the background. At the beginning of the article
Response- After extensive discussion with the authors of this manuscript, we decided to keep the paragraphs in the format they are in, so that the discussion does not lose the flow of reasoning. However, we greatly appreciate your suggestions, which were excellent for improving the manuscript.

Round 2
Reviewer 3 Report
I agree with the replies from the authors, and I accept the article in its present form,
Author Response
I appreciate the contributions